# Perceptually Optimal Color Representation of Fully Polarimetric SAR Imagery

**DOI:** 10.3390/jimaging8030067

**Published:** 2022-03-07

**Authors:** Georgia Koukiou

**Affiliations:** Electronics Laboratory, Physics Department, University of Patras, 26500 Patras, Greece; gkoukiou@upatras.gr; Tel.: +30-261-099-6147

**Keywords:** polarimetric SAR, SAR color perception, scattering mechanisms, optimal color perception, color spaces

## Abstract

The four bands of fully polarimetric SAR data convey scattering characteristics of the Earth’s background, but perceptually are not very easy for an observer to use. In this work, the four different channels of fully polarimetric SAR images, namely HH, HV, VH, and VV, are combined so that a color image of the Earth’s background is derived that is perceptually excellent for the human eye and at the same time provides accurate information regarding the scattering mechanisms in each pixel. Most of the elementary scattering mechanisms are related to specific color and land cover types. The innovative nature of the proposed approach is due to the two different consecutive coloring procedures. The first one is a fusion procedure that moves all the information contained in the four polarimetric channels into three derived RGB bands. This is achieved by means of Cholesky decomposition and brings to the RGB output the correlation properties of a natural color image. The second procedure moves the color information of the RGB image to the CIELab color space, which is perceptually uniform. The color information is then evenly distributed by means of color equalization in the CIELab color space. After that, the inverse procedure to obtain the final RGB image is performed. These two procedures bring the PolSAR information regarding the scattering mechanisms on the Earth’s surface onto a meaningful color image, the appearance of which is close to Google Earth maps. Simultaneously, they give better color correspondence to various land cover types compared with existing SAR color representation methods.

## 1. Introduction

Nowadays, radar systems are capable of providing very-high-resolution images of the Earth’s surface by means of Synthetic Aperture Radar (SAR) [1,2]. A SAR system operates in the microwave region of the electromagnetic spectrum, making it also independent of any weather effects, smoke, fog, and other related phenomena. A SAR system is capable of transmitting electromagnetic waves in two different polarizations, namely horizontal and vertical, and can receive their backscattered echoes in the same two polarizations. Each transmitting–receiving signal combination results in a single image of the same geographical scene, while the four images altogether constitute the fully polarimetric SAR image (PolSAR), which expresses the polarimetric properties of the depicted scene on Earth. They give a large amount of information regarding the scattering mechanisms on the specific pixel expressed by its scattering matrix. Several researchers, such as J. Huynen, E. Luneburg, S.R. Cloude, W.L. Cameron, and R. Touzi, have advanced [3,4,5,6,7,8,9,10,11] the theoretical framework of polarimetric decompositions, which deal with the extraction of meaningful physical properties from a SAR pixel.

It is challenging to visualize PolSAR data by color coding methods and thus extract powerful color features so as to provide additional data for a superior terrain classification. Color is one feature that is not in general considered in PolSAR visualization and classification tasks. Thus, in a survey paper [12] the performance of several features of PolSAR images, except color, is tested as far as their classification capabilities are concerned, among which texture plays an important role. Mainly, pseudo color images are formed by mapping target backscattering characteristics to different colors [13,14] or investigating and comparing different scattering parameters in various color models for visualization [15].

It is fundamental to fully understand data, especially in a complex environment where data are acquired outside the visible spectrum with totally different techniques, each one providing different kinds of information. Since the operator has no experience with the physics of the different sensors, a significant effort must be made to make data understandable by users, who often have limited technical expertise. In general, the standard appearance of PolSAR images is quite unpleasant. Despeckling is an effective operation to enhance SAR images’ appearance and improve their interpretability [16]. It is quite clear that one of the most annoying problems in handling SAR data is the grayscale display, since humans have the ability to perceive a huge amount of information with color images [17,18].

Ulhmann et al. [19] review previous attempts at PolSAR classifications using various feature combinations and then they introduce and perform an in-depth investigation of the application of color features over Pauli color-coded images besides SAR and texture features. In [20], the management of the data is proposed for providing intermediate products between the classic Level-1 and Level-2 products. These products are particularly oriented toward the end-user community. In fact, their principal characteristics are interpretability, reproducibility, and the possibility of being processed with simple algorithms widely available in the most popular software suites. Song et al. [21] proposed a deep neural network-based method to convert single-polarization grayscale SAR images to fully polarimetric images. It consists of two components: a feature extractor network to extract hierarchical multi-scale spatial features of grayscale SAR images, followed by a feature translator network to map spatial features to polarimetric features with which the polarimetric covariance matrix of each pixel can be reconstructed. The scribble-based method [22] is a semi-automatic colorization method that requires the user to scribble desired colors in certain regions. Then, the colors are propagated to the whole image based on the assumption that adjacent pixels having similar intensities should have similar colors.

In this work, we present a method for coloring PolSAR images. The method is totally different from what has been presented in the literature to date. The approach is neither based on the coloring of different textural features nor on pseudo color techniques. The proposed approach is based on a global coloring technique that originates from the challenge of taking a perceptually optimal color PolSAR image [23] when applied on multispectral images. The purpose is to obtain a color PolSAR image that is pleasant to the eye of the operator and has the appearance of physical images of the Earth’s surface as they appear from a satellite with an optical sensor (Google Earth maps). This is achieved by means of the Cholesky decomposition method [24] which provides the capability of regulating the correlation between the RGB components of the obtained color image so that it inherits the properties of a natural color image. In a second stage, the color information is transferred onto the CIELab space [25,26] where it is equalized so that it occupies all perceptible color regions. Simultaneously, the CIELab space supports perceptually equidistant colors. The reverse procedure to obtain the final RGB image follows. Simultaneously, color relevance to ordinary land cover types as well as to basic scattering mechanisms is sought.

This manuscript is organized as follows. In Section 2, the basics of polarimetry and SAR imaging are analytically presented, the elementary scattering mechanisms are analyzed, and the correlation properties of the various types of polarization images (HH, HV, VH, and VV) are explained. In Section 3, the proposed method for transforming the PolSAR bands into perceptually natural RGB color images using Cholesky decomposition is presented. In Section 4, the CIELab space is employed to perceptually equalize the color information. In Section 5, the experimental results are given that verify the elegant performance of the proposed method, while a correspondence is established among the obtained colors and the elemental scattering mechanisms that represent the various land cover types. A discussion with comparisons with existing techniques is provided in Section 6. The conclusions are drawn in Section 7.

## 2. Polarimetry and SAR Imaging

Radar images have certain features that are fundamentally different from images obtained from passive instruments using optical sensors. The image brightness levels of a scene are related to the relative strength of the microwave energy backscattered by the surface targets. The intensity of the backscattered signal varies according to the roughness, dielectric properties, and local terrain slope of the scatterers. An amplitude SAR image is a grayscale image that contains elements of visual interpretation such as tone, shape, size, pattern, texture, shadows, site, and association [2,27]. Radar polarimetry studies the way in which a radar signal interacts with a real target. This process involves the detection of any changes in the polarimetric properties of the transmitting electromagnetic wave by the target; these changes are directly linked to the scatterer’s physical and electrical properties and affect the wave polarization in different orientations (vertical–horizontal) as well as their phase differences. In this process, there are two distinct electromagnetic waves involved, the transmitted wave from the transmitting antenna and the scattered wave by the target wave that is received by the receiving antenna.

Given that the propagation of electromagnetic waves is a vectorial process, the polarimetric properties of each of the transmitted and scattered waves are directly linked to a reference system on which they are locally defined. For PolSAR imaging sensors on board an airplane or a satellite, where the transmitting and receiving antennas coincide, the monostatic configuration is as shown in Figure 1. In this configuration, called Backscatter Alignment (BSA), the backscattering matrix $S_{\left(\hat{h},\;\hat{v}\right)}$ is descriptive of the target since it describes the way in which the radar signal interacts with the target and hence it contains all the information that can be obtained from the scatterer. The backscattering matrix $S_{\left(\hat{h},\;\hat{v}\right)}$ is given by
(1)$$S_{\left(\hat{h},\;\hat{v}\right)}=\left[\begin{array}{cc} S_{\mathrm{hh}} & S_{\mathrm{hv}}\\ S_{\mathrm{vh}} & S_{\mathrm{vv}} \end{array}\right]$$
where the elements of $S_{\left(\hat{h},\;\hat{v}\right)}$ are the so-called complex scattering coefficients.

Generally, a coefficient of the backscattering matrix $S_{\left(\hat{h},\;\hat{v}\right)}$ expresses the change in the polarization of the transmitted wave caused by the target in a given direction that is received in either direction of the antenna. These coefficients depend on the physical, electrical, and geometrical properties of targets as well as on their relative orientation with respect to the antenna. The *S*_hv_ and *S*_vh_ coefficients of $S_{\left(\hat{h},\;\hat{v}\right)}$ are the cross-polar polarization combinations. In the monostatic configuration, the off-diagonal coefficients are equal ($S_{\mathrm{hv}}=S_{\mathrm{vh}}$), since the reciprocity theorem applies [28]. Accordingly, $S_{\left(\hat{h},\;\hat{v}\right)}\;$has three independent complex parameters.

Elementary scattering mechanisms [29] correspond to ideal lossless isotropic targets whose backscattering behavior is well known. These scatterers are divided into two broad categories, symmetric and asymmetric elementary scatterers. The one-dimensional scatterer is the dipole. A two-dimensional scatterer is the plane scatterer (or trihedral or spherical scatterer; these scatterers present different geometric compositions, but they share the same scattering properties), which is related to odd-bounce scattering mechanisms. One three-dimensional scatterer is the quarter-phase depolarizer that causes a phase difference equal to ±π/2 in the vertically polarized component of the incident wave. Another three-dimensional scatterer is the diplane scatterer (or corner reflector or diplane) that is related to even-bounce scattering mechanisms. The asymmetric scattering mechanisms are the general circular depolarizers, the left and right helix.

The plane scatterer does not alter the incident polarizations of the wave. The most important feature of the plane scatterer is that it remains invariant to rotations around the radar’s Line of Sight (LOS). Knowledge of the behavior of the elementary scattering mechanisms by means of their scattering matrix can give insight into the scattering behavior of each SAR pixel depending on the scattering matrix of the specific pixel. The backscattering matrices of the elementary scatterers are:(2)$$\mathrm{horizontally}\;\mathrm{aligned}\;\mathrm{dipole}\;S_{\left(\hat{h},\;\hat{v}\right)}=\left[\;\begin{array}{cc} 1 & 0\\ 0 & 0 \end{array}\;\right]$$
(3)$$\mathrm{plane}\;\mathrm{scatterer}\;S_{\left(\hat{h},\;\hat{v}\right)}=\left[\;\begin{array}{cc} 1 & 0\\ 0 & 1 \end{array}\;\right]\;$$
(4)$$\mathrm{horizontally}\;\mathrm{oriented}\;\mathrm{quarter}\;\mathrm{phase}\;\mathrm{depolarizer}\;S_{\left(\hat{h},\;\hat{v}\right)}^{\mathrm{QWD}}=\left[\;\begin{array}{cc} 1 & 0\\ 0 & \pm i \end{array}\;\right]$$
(5)$$\mathrm{diplane}\;\mathrm{scatterer}\;S_{\left(\hat{h},\;\hat{v}\right)}=\left[\;\begin{array}{cc} 1 & 0\\ 0 & –1 \end{array}\;\right]$$
(6)$$\mathrm{left}\;\mathrm{helix},\;\mathrm{arbitrarily}\;\mathrm{oriented}\;S_{\left(\hat{h},\;\hat{v}\right)}=\frac{e^{–2i\varphi }}{2}\left[\;\begin{array}{cc} 1 & i\\ i & –1 \end{array}\;\right]$$
(7)$$\mathrm{right}\;\mathrm{helix},\;\mathrm{arbitrarily}\;\mathrm{oriented}\;S_{\left(\hat{h},\;\hat{v}\right)}=\frac{e^{–2i\varphi }}{2}\left[\;\begin{array}{cc} 1 & –i\\ –i & –1 \end{array}\;\right]\;$$

The horizontally oriented quarter-phase depolarizer changes the phase of the vertically polarized component of the incident wave by 90° while leaving unchanged the horizontally polarized component. The diplane scatterer changes the phase of the vertically polarized component of the incident wave by 180° while leaving unchanged the horizontally polarized component.

In Figure 2 are presented amplitude SAR images with a high spatial resolution. The scene comes from Vancouver in British Columbia, Canada. The data are fully polarimetric SAR data acquired by the RADAR-SAT 2 satellite (SLC data). The Sentinel Application Platform (SNAP) from ESA was employed to geocode the data. The figure depicts the amplitude images resulting from all combinations of SAR polarizations (HH, HV, VH, and VV).

The four SAR images convey information that is not totally different. A way to estimate the common information among the images is to evaluate their covariance matrix or the corresponding correlation coefficient matrix [30]. The correlation coefficient matrix is given in Table 1, while the covariance matrix is given in Table 2. From the two matrices, the following can be deduced:From the correlation coefficient matrix, a very large correlation (0.96) is found between channels VH and HV, while a significant correlation is noted among channels HH and VV;From the covariance matrix is observed significant energy in channels HH and VV; andFrom the covariance matrix is observed significant common energy between HH and VV.

**Table 1 jimaging-08-00067-t001:** Correlation coefficient matrix of the SAR amplitude images of Figure 2.

	HH	HV	VH	VV
**HH**	1	0.36	0.36	0.62
**HV**	0.36	1	0.97	0.33
**VH**	0.36	0.97	1	0.33
**VV**	0.62	0.33	0.33	1

**Table 2 jimaging-08-00067-t002:** Covariance matrix of the SAR amplitude images of Figure 2.

	HH	HV	VH	VV
**HH**	2.61 × 10^6^	0.34 × 10^6^	0.34 × 10^6^	1.49 × 10^6^
**HV**	0.34 × 10^6^	0.35 × 10^6^	0.35 × 10^6^	0.28 × 10^6^
**VH**	0.34 × 10^6^	0.33 × 10^6^	0.35 × 10^6^	0.28 × 10^6^
**VV**	1.49 × 10^6^	0.28 × 10^6^	0.28 × 10^6^	2.11 × 10^6^

The four amplitude SAR images that are depicted in Figure 2 cannot be used to straightforwardly create an acceptable RGB image. Although the HV and VH amplitude images are highly correlated (the correlation coefficient equals 0.97) and, consequently, one of them can be neglected, if the rest of the images are used to represent respectively the R, G, and B components of a color image, the obtained result will be discouraging as is shown in Figure 3. This was expected since the information content of the HV amplitude image, which corresponds to the Green (G) component, is quite poor, resulting in a red-purplish color appearance.

## 3. Fusion of the Four PolSAR Images into a Natural RGB Representation

In this work, we apply, as a first stage, an attractive method to combine the SAR images depicted in Figure 2 to derive a color image with meaningful information regarding various land cover types. The information contained in the four combinations of transmitted/received SAR images, i.e., HH, HV, VH, and VV, is gathered into an RGB image based on the fact that the derived image should be pleasing to the human eye and the maximum color information is perceived by the human vision system. In order to achieve this goal, the three derived R, G, and B color components must possess the same correlation properties as the natural color images. In this way, the characteristics of human color perception are embodied in the proposed transformation procedure. The goal of the proposed method is not to totally decorrelate the data, but to control the correlation between the color components of the final image. Orthogonal components, i.e., those obtained by PCA transformation, are not suitable for color representation since the image energy is not distributed evenly among the color components; instead, most of the energy lies along the principal component [31].

Color perception is based on the activity of cones, called S cones, M cones, and L cones (with peak sensitivity at short, medium, and long wavelengths, respectively). The color matching functions for the primaries of the RGB system are depicted in Figure 4. This is a consequence of the overlapping sensitivity curves of the different types of cones in the human eye [26] (p. 105) as well as the color matching functions for the primary colors of the RGB system given in Figure 4.

Various approaches can be found in the literature for pixel-level fusion [23,32]. The great variety of image fusion methods can be justified by the complexity of the problem, the different types of data involved, and the different aims of each application. Fusion can be employed to provide improved visual interpretation by means of combining different spectral characteristics or image modalities. Pixel-level fusion techniques can also be used to improve the efficiency of classification and detection algorithms. The proposed linear transformation is a pixel-level fusion approach suitable for improving perceptual vision. The required linear transformation is of the form
(8)$$y=A^{T}x$$
where x and y the vector representations of the initial and final images, respectively. Their covariance matrices are related as follows
(9)$$C_{y}=A^{T}C_{x}A$$

The required values of the elements of the final covariance matrix $C_{y}$ are based on the study of natural color images. The user’s selection of the degree of correlation between the bands of the final color image depends on the application. The employment of a covariance matrix $C_{y},$ similar to those of natural color images, ensures that the final color image will be easily interpreted by the human eye.

The matrices $C_{x}$ and $C_{y}$ are of the same dimensionality while the transformation matrix A can be evaluated by means of Cholesky factorization. According to this method, a positive definite matrix can be analyzed by means of an upper triangular matrix Q so that
(10)$$S=Q^{T}Q$$

The matrices $C_{x}$ and $C_{y}$ based on Cholesky factorization can be written as
(11)$$C_{x}=Q_{x}^{T}Q_{x}$$
(12)$$C_{y}=Q_{y}^{T}Q_{y}$$
and by combining (9)–(12), the required transformation matrix A is obtained as
(13)$$A=Q_{x}^{-1}Q_{y}$$

The final form of the transformation matrix A reveals that the proposed transformation depends on the statistical properties of the initial set of PolSAR data and simultaneously the statistical properties of the natural color images are taken into consideration as well through the matrix $Q_{y}$. The final vector of y is of the same dimensionality as the initial vector x. However, the total information is transferred to the first three components that will contribute to the final RGB color image.

The evaluation of the desired covariance matrix $C_{y}$ is based on the statistical properties of the natural color images but it can also incorporate requirements posed by the specific application or the operator who is going to use the final image. The relation between the covariance matrix $C_{y}$ and the correlation coefficient matrix $R_{y}$ is
(14)$$C_{y}=\Sigma R_{y}\Sigma^{Τ}$$
where
(15)$$\Sigma =\left[\begin{array}{c} \begin{array}{cc} \sigma_{y1} & 0\\ 0 & \sigma_{y2} \end{array}\;\;\begin{array}{cc} \ldots & 0\\ \ldots & 0 \end{array}\\ \;\;\;\begin{array}{cc} \;\;\;\vdots & \\ \;\;\;0 & \;\;0 \end{array}\;\;\;\;\;\begin{array}{cc} \;\ldots & \vdots \\ \ldots & \sigma_{yK} \end{array} \end{array}\right]$$
is the matrix with the variances in the new vectors on the main diagonals, and
(16)$$R_{y}=\left[\begin{array}{c} \begin{array}{cc} 1 & r_{R,G}\\ r_{G,R} & 1 \end{array}\;\;\begin{array}{cc} \ldots & 0\\ \ldots & 0 \end{array}\\ \;\;\;\begin{array}{cc} \;\;\;\vdots & \\ \;\;\;0 & \;\;0 \end{array}\;\;\;\;\;\begin{array}{cc} \;\ldots & \vdots \\ \ldots & 1 \end{array} \end{array}\right]$$
is the matrix with the desired correlation coefficients.

The steps of the proposed algorithm employed to transform the information in the PolSAR data into an RGB image can be outlined as follows:Determine the desired matrix $R_{y}$ (16) and evaluate the corresponding matrix $C_{y}$ by means of (14);Evaluate the matrix $C_{x}$ from the initial PolSAR images;Evaluate matrices $Q_{x}$ and $Q_{y}$ from $C_{x}$ and $C_{y}$ by means of Cholesky factorization; andConstruct the transformation matrix A by means of (13).

The next stage of the proposed method is to equalize the color information in the RGB image by means of the CIELab color space.

## 4. Equalization of Color Information in the CIELab Color Space

The CIELab space is considered to be a perceptually uniform color space and is employed in this work to enhance the perceptibility of the synthesized RGB SAR information. This is very important for the derived colored SAR images since perceptually equal small differences between various colors are depicted as equidistant in the CIELab space. By employing the CIELab color space, it becomes possible to utilize all available colors by equalizing the information on the L, a, and b axes.

The CIELab color system was standardized as a uniform color space by the Commission Internationale de l’Eclairage (CIE) [25,26] and supports color management in order to assess realizable color information with high perceptibility. It contains three dimensions, one of which is achromatic and corresponds to the Luminance L, while the other two directions a and b are used for representing color. L ranges from 0, which corresponds to black, to 100 for perfect white. The positive a axis represents the amount of purplish red, while the negative a axis the amount of green. The axis b represents the yellow in the positive direction while the blue is represented by the negative direction. The maximum values on the axes a and b are a function of L and range from −200 to 200. The maximum and minimum values of a and b are those resulting in acceptable RGB values after conversion.

The conversion of the RGB color space to the CIELab space and vice versa is achieved with an intermediate step in the CIE XYZ space:(17)$$\left[\begin{array}{c} X\\ Y\\ Z \end{array}\right]=\frac{1}{0.17697}\left[\begin{array}{ccc} 0.490 & 0.310 & 0.200\\ 0.177 & 0.812 & 0.010\\ 0.000 & 0.010 & 0.990 \end{array}\right]\left[\begin{array}{c} R\\ G\\ B \end{array}\right]$$
(18)$$L=116\;f\left(\frac{Y}{Y_{n}}\right)-16\;a=500\left[f\left(\frac{X}{X_{n}}\right)-f\left(\frac{Y}{Y_{n}}\right)\right]\;b=200\left[f\left(\frac{Y}{Y_{n}}\right)-f\left(\frac{Z}{Z_{n}}\right)\right]$$
where *f*(*x*) is
(19)fx=x13                  x>0.0088567.787x+16116x ≤0.008856

All relevant relationships can be found in any colorimetry book [25,26].

The transformation from the RGB color space to the CIELab space based on (17) to (19), equalization in CIELab, and transformation back to RGB were performed by means of MATLAB from MathWorks.

## 5. Experimental Approach and Color Relevance to Scattering Mechanisms

According to the PolSAR coloring approach proposed in this work, two consecutive coloring stages are performed in order to obtain a perceptually rich image and simultaneously present a consistent correspondence between specific scattering mechanisms and the color palettes.

In the first stage, knowledge of the correlation properties of natural color images is required. For this purpose, we examined a variety of natural color images and tested correlation properties among RGB color components. For demonstration purposes, three different RGB images are presented in Figure 5 along with their correlation coefficient matrix. Commenting on these images, we have to note that the image “Butterflies” presents an expected correlation among the different color components. The image “Candies” presents almost zero correlation among Red and Blue, and this is an indication that the image is not close to a natural one since it lacks the coexistence of Red and Blue. Finally, the Google Earth image presents totally correlated color components. This is not an image that is pleasing to the human eye since, according to the author’s perception, the coexistence of all color components in the same degree and on a very dense scale gives a flat sense for the color palette, since there is a lack of abrupt alternations between the color components. This is not the same for the other two images in Figure 5. Thus, the image “Candies” has large blocks of the same color that are neighbors only in the boundaries of the blocks, while the “Butterflies” image, being a quite natural image, has small and simultaneously large color regions with a large number of neighboring color regions. The increase in the neighboring of different color regions increases the correlation coefficients.

Based on the correlation coefficients of the image “Butterflies” as well as a large number of other natural images and especially the correlation between colors imposed by the curves in Figure 4, we decided to apply in the first stage of our method the correlation coefficients matrix given in Table 3 in order to create an RGB SAR image according to Section 4. The obtained result is shown in Figure 6. This RGB image presents a natural appearance very close to Google Earth images (compared with the image in Figure 5c). However, the derived RGB image presents various regions, such as the river or the residential area, whose features have more intense colors due to the different scattering mechanisms of these regions.

From the image in Figure 6 when compared with the Google Earth image in Figure 5c, we can observe that the residential area on the top of the image is clearly separated from the rest of the land cover types. The same is evident for the industrial area at the two sides of the wide river, as well as for the grass and cultivated areas depicted all over the region covered by the image.

The image in Figure 6, obtained by means of fusing the SAR amplitude bands so that the specific correlation coefficient matrix in Table 3 is established among its color components, was subsequently transformed into the CIELab color space for equalization.

In the CIELAB color space, the specific image presents the following range for the L, a, and b values:L: 2 to 100 and a bell-shaped histogram;a: −82 to 77 and a bell-shaped histogram (from green to purplish red); andb: −106 to 94 and a bell-shaped histogram (from blue to yellow).

The three histograms were equalized using the simple MATLAB equalization routine. Regarding the new equalized L, a, and b histograms, although they cover almost the same range for their values, their curves tend to become uniform, that is:L: 0 to 100, an almost uniformly shaped histogram;a: −78 to 112, an almost uniformly shaped histogram (from green to purplish red); andb: −127 to 113, an almost uniformly shaped histogram (from blue to yellow).

The equalized L, a, and b components are transformed back to the RGB color space for final use. The obtained image is the one shown in Figure 7. This color image is obviously richer in colors to the human eye, and details of the background are more distinct comparatively to those in Figure 6 or the simple black and white images in Figure 2. A more detailed correspondence between colors and scattering mechanisms is provided in the following. For this purpose, we focused on three different areas of the big picture as shown in Figure 8 and comment on the scatterer–color correspondence.

To quantitively compare the images in Figure 6 and Figure 7, the histograms of the images were evaluated. Accordingly, in Figure 8a are given the histograms of the components in the CIELab space before equalization corresponding to the image in Figure 6, while in Figure 8b are given the histograms of the components of the CIELab space after equalization corresponding to the image in Figure 7. It is obvious that the equalized L-component (left), a-component (center), and b-component (right) have a larger number of pixels at the edges of the equalized histograms, giving the equalized image a richer appearance.

Our experimental procedure was thorough and matched specific colors of the final images (Figure 7 and Figure 9) with both the land cover types and the elementary scattering mechanisms. Accordingly, we obtained the following results:The Water area is represented with 73% by the trihedral as a primary scattering mechanism. The trihedral gives a reddish color to the SAR cells that contain this scatterer;The Urban area corresponds 29% to the quarter-wave device scattering mechanism, 21% to the dipole, 16% to the narrow diplane, and 15% to the cylinder scattering mechanism. Due to this variety of scatterer types in the urban area, almost all colors given in Figure 7 and Figure 8a have a multicolor appearance.The Forest area corresponds 29% to the cylinder scattering mechanism, 26% to the quarter-wave device scattering mechanism, and 17% to the dipole scatterer. The two main scattering mechanisms, i.e., the cylinder and the quarter-wave device, provide the final image with a green or dark green color.The Agriculture area is mostly represented by the trihedral scatterer (with 52%) and the cylinder scatterer (with 31%). The color that is obtained by the two scatterers is the green or light green.

**Figure 9 jimaging-08-00067-f009:**
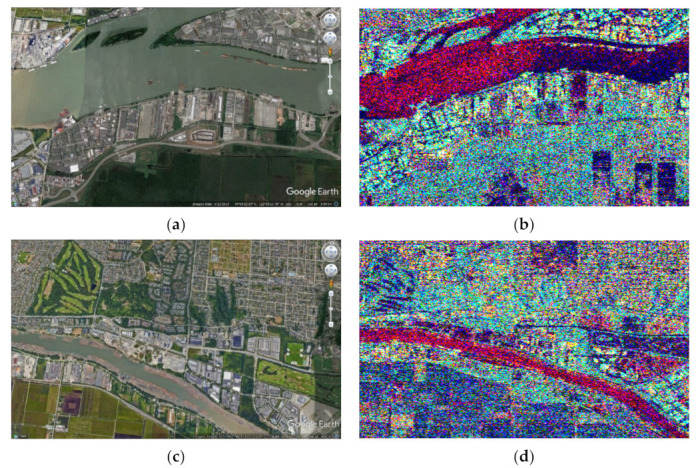
Three different regions from the wider area of Vancouver from Google Earth maps (**a**,**c**,**e**) and their corresponding representation (**b**,**d**,**f**) after the two-stage procedure proposed in this work. Agriculture and Forest areas appear as green or dark green, and the Urban area appears in multiple colors due to the large variety of scattering mechanisms they contain. Bare land appears reddish like the smooth surface of the water where the trihedral scatterer dominates.

A detailed representation of the contribution of each scattering mechanism in the four basic land cover types is given in Table 4 [33]. The region with the “smoothest” scattering surface, the Water area, has a reddish color.

## 6. Discussion and Comparisons

The proposed PolSAR coloring technique is novel as far as its completeness in terms of the elements it takes into account for the coloring is concerned, such as the correlation between the R, G, and B components that resembles that of the natural color images and simultaneously covers all perceptible color shades by means of the CIELab color space. The proposed PolSAR coloring technique has advantages over existing approaches, which are explained below.

Mainly, pseudo color images are formed by mapping target backscattering characteristics to different colors [13,14] or investigating and comparing different scattering parameters in various color models for visualization [15]. In particular, the work in [13] performs colorization in the CIELab space without moving all of the information into a first-stage RGB image with specific correlation coefficients between the R, G, and B components. In [14], a pseudo coloring technique was proposed for SAR images by means of the CIELab color space. No correspondence between the various colors and types of scatterers was established.

In [19], the authors review previous attempts at PolSAR classifications using various feature combinations and apply pseudo color imaging on a Pauli H and V basis, providing further improvements in terms of class discrimination. The technique is restricted to a specific scattering basis and is not global.

In [20], the management of the data is proposed for providing intermediate products between the classic Level-1 and Level-2 products. These products are particularly oriented toward the end-user community. In fact, their principal characteristics are interpretability, reproducibility, and the possibility of being processed with simple algorithms widely available in the most popular software suites. The obtained coloring results are perceptually inferior to those produced with the technique proposed in the present work.

In [21], a deep neural network-based method is proposed to convert single-polarization grayscale SAR images to fully polarimetric images. It consists of two components: a feature extractor network to extract hierarchical multi-scale spatial features of grayscale SAR images, followed by a feature translator network to map spatial features to polarimetric features with which the polarimetric covariance matrix of each pixel can be reconstructed. Compared with the method proposed in this work, the approach of [21] is quite blind since it maps spatial features to polarimetric ones.

In [34], a method is proposed to colorize SAR images using a multidomain cycle-consistent generative adversarial network (MC-GAN). The approach uses land cover templates to match various colors based on information from visual images. Various existing coloring platforms are used for this purpose. No serious mathematical background is presented.

In conclusion, the PolSAR coloring method proposed in this work follows a complete mathematical model in order to correlate color bands according to natural color images and simultaneously equalize color bands for maximum perceptibility. Additionally, a specific correspondence is established between the colors in the derived PolSAR color images and the elementary scattering mechanisms.

## 7. Conclusions

In this work, the information contained in the four amplitude bands of PolSAR images was transferred to a meaningful color RGB image that establishes a color correspondence to physical scattering mechanisms and land cover types. A strict mathematical process based on Cholesky decomposition was applied to determine the color correlation properties of the created RGB image and, subsequently, the CIELab space was employed to perceptually enhance the color palette of the final RGB image.

This two-phase procedure brings the PolSAR information regarding the scattering mechanisms on the Earth’s surface onto a meaningful color image, the appearance of which is close to Google Earth maps and simultaneously gives better color correspondence to various land cover types compared with existing SAR color representation methods.

The most important advantage of the proposed method is that it establishes a significant correspondence between the color shades and the elemental scattering mechanisms, giving the user the ability to distinguish based on the color the way each land cover type reflects the electromagnetic energy.

## Figures and Tables

**Figure 1 jimaging-08-00067-f001:**
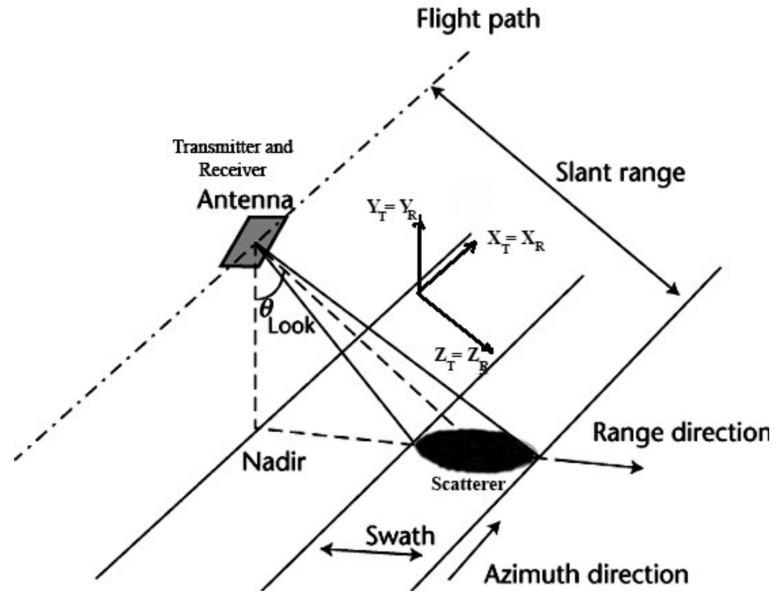
Polarimetric scattering process for the monostatic configuration, where the vectors are defined with respect to the radar antenna (BSA, Backscatter Alignment).

**Figure 2 jimaging-08-00067-f002:**
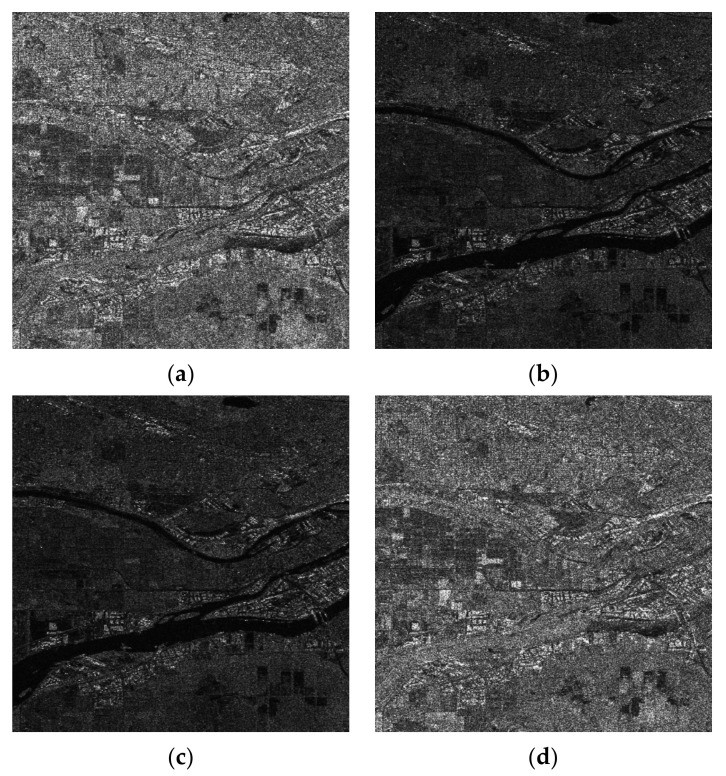
Amplitude SAR images with a high spatial resolution. The scene comes from Vancouver in British Columbia, Canada and was acquired by the fully polarimetric SAR sensor on board RADAR-SAT 2. The figure depicts the images resulting from all combinations of SAR polarizations (HH, HV, VH, and VV). (**a**) HH; (**b**) HV; (**c**) VH; (**d**) VV.

**Figure 3 jimaging-08-00067-f003:**
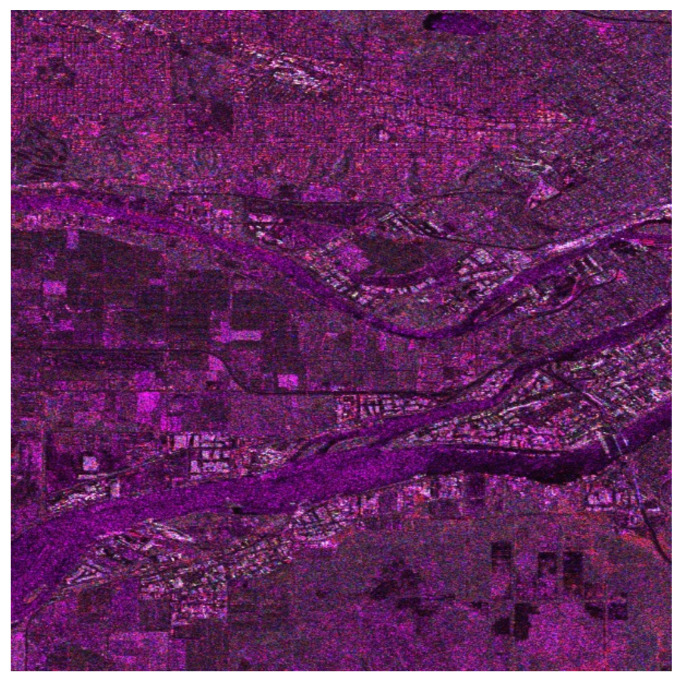
The RGB image obtained using the HH, HV, and VV amplitude SAR images for the R, G, and B components, respectively. The color appearance is poor due to the low energy in the HV-green component.

**Figure 4 jimaging-08-00067-f004:**
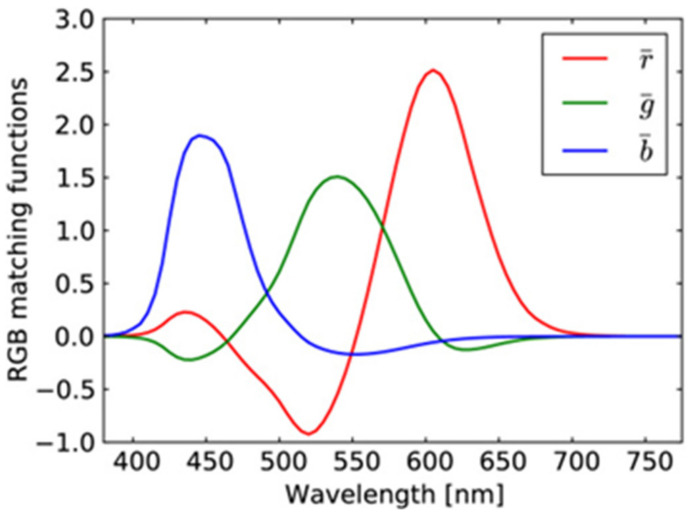
Color matching functions for the primaries of the RGB system.

**Figure 5 jimaging-08-00067-f005:**
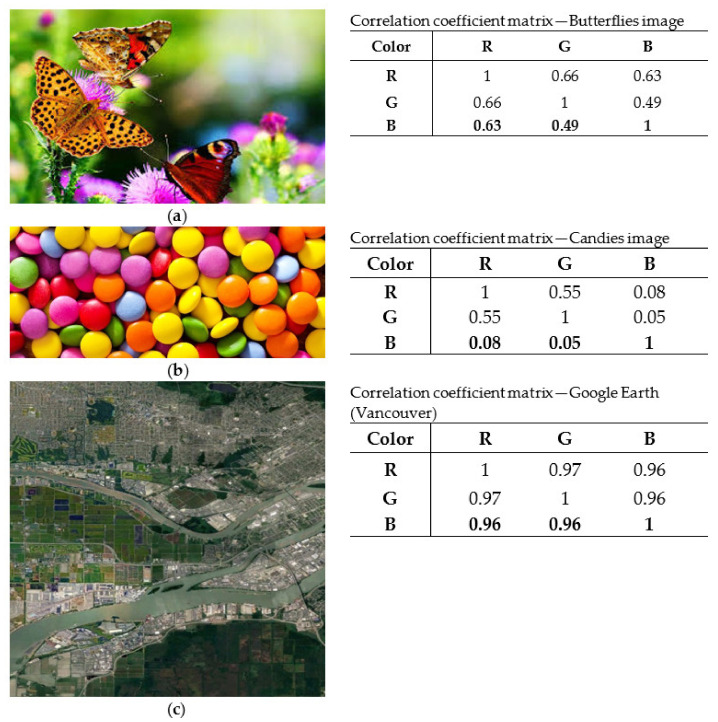
Three color images. (**a**) The butterflies can be considered as being a natural image. (**b**,**c**) can be considered natural images with a special formation of the correlation coefficient matrix.

**Figure 6 jimaging-08-00067-f006:**
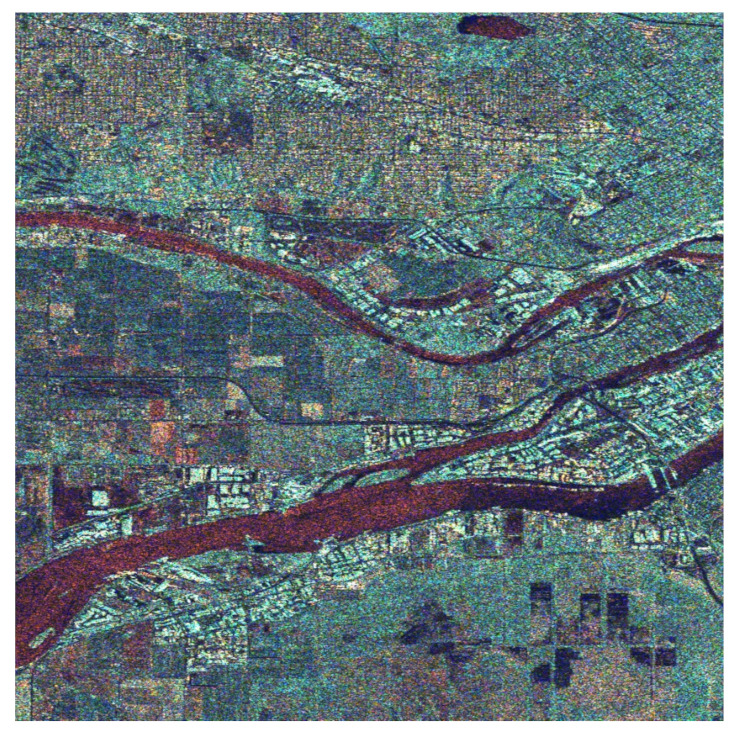
RGB image of the Vancouver region obtained after the transformation (8), which is a fusion algorithm based on Cholesky decomposition and correlation of the RGB components close to that of the natural images.

**Figure 7 jimaging-08-00067-f007:**
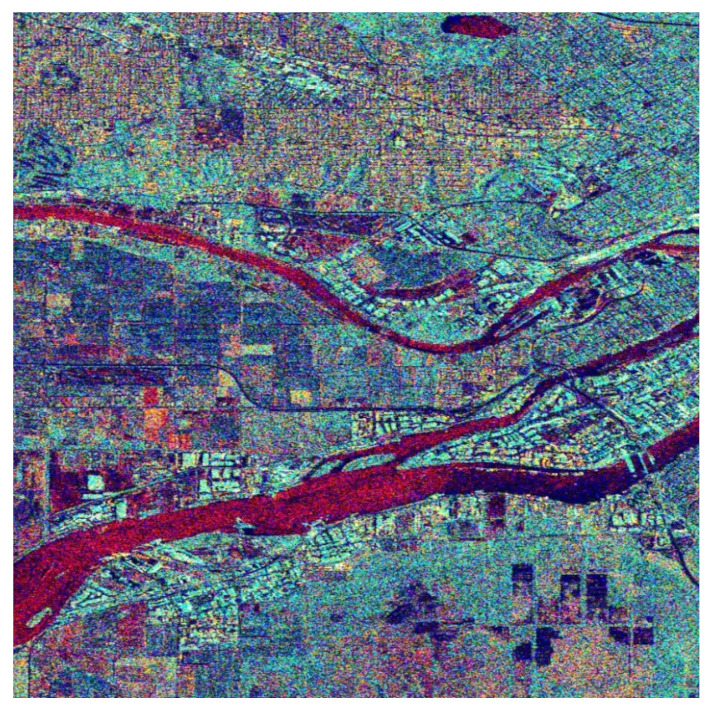
RGB image of the Vancouver region obtained after applying equalization in the CIELab color space of the color information of the RGB image in Figure 6.

**Figure 8 jimaging-08-00067-f008:**
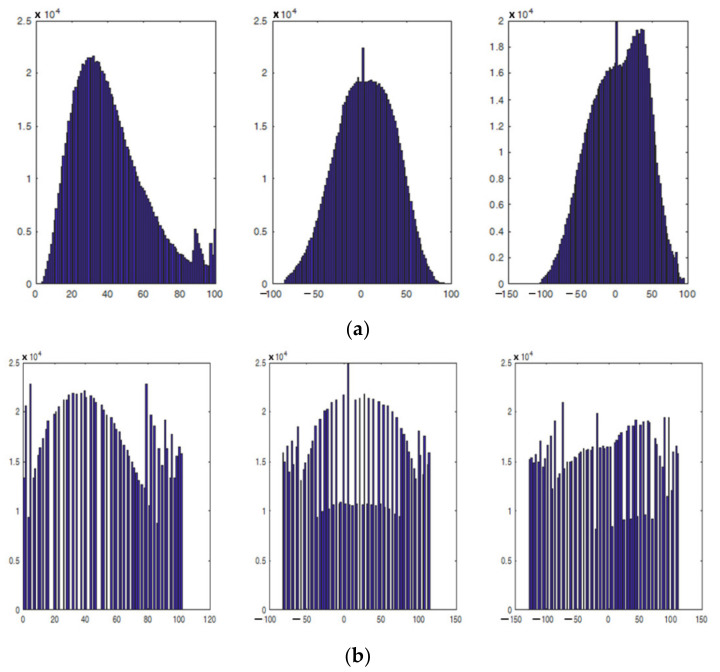
Histograms in the CIELab space (**a**) before equalization corresponding to the image in Figure 6 and (**b**) after equalization corresponding to the image in Figure 7. It is obvious that the equalized L-component (left), a-component (center), and b-component (right) have a larger number of pixels at the beginning and the end of the histograms.

**Table 3 jimaging-08-00067-t003:** Correlation coefficient matrix applied through Cholesky decomposition to create the RGB SAR imagery.

Color	R	G	B
**R**	1	0.66	0.33
**G**	0.66	1	0.66
**B**	0.33	0.66	1

**Table 4 jimaging-08-00067-t004:** Observation probability for each of the eight scattering mechanisms for each land cover type.

	Trihedral	Dihedral	Dipole	Cylinder	Narrow Diplane	¼ Wave Device	Left Helix	Right Helix
Water Area	0.7310	0.0016	0.0183	0.2073	0.0048	0.036	0.0007	0.0005
Urban Area	0.0288	0.0649	0.2137	0.1450	0.1601	0.2869	0.0604	0.0401
Forest Area	0.1066	0.0246	0.1719	0.2924	0.0753	0.2603	0.0394	0.0295
Agriculture Area	0.5199	0.0030	0.0481	0.3120	0.0129	0.0929	0.0071	0.0042

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
