# Peer review of "Perceptually Optimal Color Representation of Fully Polarimetric SAR Imagery"

_2313-433X, 2022, doi:10.3390/jimaging8030067_

Round 1
Reviewer 1 Report
See the attachment

Reviewer 2 Report
- The abstract logic is chaotic and the cohesion is not smooth. Such as “the first one is a fusion procedure”, “the second procedure”, “this two-phase procedure”.
- The full text uses a large number of long sentences, resulting in poor readability of the paper.
- The equation is part of the sentence, and punctuation should be added at the end of the equations.
- There are still grammatical errors in the text, such as section 1 should be changed to Section 2, etc.
- There are problems in the layout of the full text pictures and tables, such as the use of unclear screenshots in the pictures, the layout of the table beyond the boundary, the layout of Table 4 and Figure 8 overlapping, and so on.
- There are too few references. It is recommended to cite more recent research results in the past three years.
- My main concern is that compared with existing SAR color representation methods, the proposed method is competitive, but compared with the state-of-the-art methods in the past three years, the competitiveness of the proposed method is uncertain. Therefore, it is necessary to suggest adding the results of recent years to evaluate the performance of the proposed method.
Round 2
Reviewer 2 Report
Thanks for your response. My concerns are all addressed. I recommend acceptance of this manuscript.